# Gene Duplication and Functional Diversification of MADS-Box Genes in *Malus × domestica* following WGD: Implications for Fruit Type and Floral Organ Evolution

**DOI:** 10.3390/ijms25168962

**Published:** 2024-08-17

**Authors:** Baoan Wang, Yao Xiao, Mengbo Yan, Wenqi Fan, Yuandi Zhu, Wei Li, Tianzhong Li

**Affiliations:** College of Horticulture, China Agricultural University, Beijing 100193, China; poanwang@cau.edu.cn (B.W.); xiaoyao6703@sina.com (Y.X.); ymb215@cau.edu.cn (M.Y.); fanwenqi639@163.com (W.F.); zhuyd@cau.edu.cn (Y.Z.); liwei_613@cau.edu.cn (W.L.)

**Keywords:** duplication, functional differentiation, MADS-box, fruit types, floral organs, evolution

## Abstract

The evolution of the MADS-box gene family is essential for the rapid differentiation of floral organs and fruit types in angiosperms. Two key processes drive the evolution of gene families: gene duplication and functional differentiation. Duplicated copies provide the material for variation, while advantageous mutations can confer new functions on gene copies. In this study, we selected the Rosaceae family, which includes a variety of fruit types and flower organs, as well as species that existed before and after whole-genome duplication (WGD). The results indicate that different fruit types are associated with different copies of MADS-box gene family duplications and WGD events. While most gene copies derived from WGD have been lost, MADS-box genes not only retain copies derived from WGD but also undergo further gene duplication. The sequences, protein structures, and expression patterns of these gene copies have undergone significant differentiation. This work provides a clear example of MADS-box genes in the context of gene duplication and functional differentiation, offering new insights into the evolution of fruit types and floral organs.

## 1. Introduction

Whole-genome duplication (WGD) events have been crucial in the adaptive evolution and diversification of angiosperms [1,2,3]. Approximately 100 to 120 million years ago (Ma), several ancient polyploidy events occurred in angiosperms [4], with four significant events detected in major clades: a whole-genome triplication in the common ancestor of core eudicots (γ) [5] and WGDs in monocots (τ) [6], magnoliids (λ) [7], and the Nymphaeales lineage (π) [8]. Numerous species-rich families provide strong evidence of WGD, with recent polyploidizations identified in *Arabidopsis thaliana* [9], *Populus trichocarpa* [10], and *Glycine max* [11]. The Maleae tribe within the Rosaceae family has undergone a recent WGD event [12], leading to the development of pome fruit. Studying WGD’s role in the evolution of diverse fruit types is highly valuable.

The Rosaceae family comprises over 3000 species [13]. Recent molecular phylogenetic analyses have classified Rosaceae species into three subfamilies, Dryadoideae, Rosoideae, and Amygdaloideae, with Maleae belonging to Amygdaloideae [14]. Both Amygdaloideae and Rosoideae include numerous species that are vital for human nutrition and ornamental purposes. These subfamilies exhibit wide phenotypic diversity in fruit types, such as the fleshy fruit known as a pome (found in apples and pears), which is formed from the fusion of the hypanthium and ovary; the drupe (seen in peaches and cherries), a fleshy fruit with a hard endocarp enclosing a seed; and the achenetum and drupetum (typical of strawberries and raspberries), which are fleshy fruits derived from the receptacle and/or multiple achenes [15,16].

Generally, fruits develop from floral tissue following pollination; thus, the diversity of fruits is closely linked to the structural variations in floral organs [17]. Both apple and peach flowers possess a hypanthium, a cup-like structure formed by the fusion of the lower portions of the sepals, petals, and stamens [18,19]. However, in peaches, the fruit is formed solely from the ovary. Strawberry and raspberry flowers feature raised stem tips known as receptacles, which connect to the bases of numerous carpels. Each carpel has the potential to develop into a single small, one-seeded fruit after fertilization [15,20,21]. These structural differences in floral organs directly influence the physiological processes of fruit development.

The MADS-box family genes are well known for their role in regulating the morphogenesis of floral organs [22]. According to the latest model of flower development, the identity of floral organs is determined by the combinatorial actions of five classes of MADS-box genes, named A, B, C, D, and E, most of which belong to the type II group (also known as the MIKC type) [23,24]. Specifically, class A genes include *APETALA1* (*AP1*) and *AP2*; class B genes include *AP3 and PISTILLATA* (*PI*); class C genes include *AGAMOUS* (*AG*); class D genes include *SEEDSTICK* (*STK*), *SHATTERPROOF1* (*SHP1*), and *SHP2*; and class E genes include *SEPALLATA1* (*SEP1*), *SEP2*, and *SEP3* [25,26,27,28,29]. Class A and E genes determine sepal formation; class A, B, and E genes determine petal formation; class B, C, and E genes determine stamen formation; class C and E genes determine carpel formation; and class C, D, and E genes control ovule formation and development [30,31].

The MADS-box family genes are widely present in angiosperms and have a large number of members, many of which originated from duplications of a common ancestor gene [22,32,33]. For instance, following the γ event, multiple copies of the *AP1*, *AP3*, *AG*, *SEP1*, *AGL6*, and *SOC1* genes remain [34]. After genome polyploidy, the retention of priority genes and subsequent divergence have contributed to the development of new traits. There has been a significant expansion in the number of MADS-box genes in the *Malus* and *Pyrus* genera, comprising 132 and 74 genes, respectively [35,36]. In contrast, the *Prunus* genus, which did not experience the recent WGD, has an average of 37 MADS-box genes [37,38,39]. Although there is evidence indicating the expression of duplicated MADS-box genes in plant tissue [35,36], it is still unclear how this leads to the evolution of the pome. Further analysis is needed to elucidate the functional differentiation after WGD in Maleae.

In this study, we reconstructed the phylogenetic tree and the ancestral Rosaceae karyotype (ARK) with nine ancient chromosomes, encompassing the Rosoideae subfamily, the *Prunus* genus, and the Maleae tribe, both before and after the WGD. Karyotype evolution analysis revealed that the Maleae tribe has continuously experienced chromosome breakage and rearrangement following the WGD, and it was inferred that this event occurred at approximately 14.69 Ma. We found that approximately 30.71% of the gene copies derived from the WGD are still expressed in plant tissue. Among these, the orthologous groups of MADS18 and MADS6 contain both WGD and single gene duplication copies. These copies undergo sequence differential evolution, affecting their physical and chemical properties and protein structure, and exhibit significant differential expression in apple fruits at different developmental stages and in different floral organs, providing new insights into the evolution of fruit types and floral organs.

## 2. Results

### 2.1. The Expansion in the Number of MADS-Box Genes across Different Fruit Types Is Correlated with WGD in Rosaceae

To investigate the role of the recent whole-genome duplication (WGD) in the evolution of pome fruit, we selected four species from the Maleae tribe, which experienced the recent WGD event, and eight species from the *Prunus* genus and the Rosoideae subfamily, which did not undergo the recent WGD event, to perform a comparative genomic analysis (Appendix A).

*Crataegus pinnatifida*, *Eriobotrya japonica*, *Pyrus pyrifolia*, and *Malus × domestica* of the Maleae tribe are all pome fruits with a chromosome number of x = 17. *Prunus avium*, *Prunus persica*, *Prunus armeniaca*, and *Prunus salicina* in the *Prunus* genus are all drupe fruits with a chromosome number of x = 8. *Fragaria* × *ananassa*, *Rubus argutus*, *Rubus occidentalis*, and *Rubus idaeus* in the Rosoideae are all drupetum/achenetum fruits with a chromosome number of x = 7 (Figure 1A). The chromosome number in the Maleae tribe is about twice that of the *Prunus* genus and the Rosoideae.

The crown age of the Rosaceae family is estimated to be 94.46–96.36 million years (Ma) [14]. Combined with phylogenetic analysis, we estimate that Maleae and *Prunus* diverged between 40.10 and 81.09 Ma, and the most recent common ancestor (MRCA) of Maleae emerged between 6.44 and 12.36 Ma. The MRCA of *Prunus* emerged between 6.86 and 23.04 Ma, and the MRCA of Rosoideae emerged between 26.69 and 66.67 Ma (Figure 1A). We define the MRCA of Maleae as the P (pome) node based on the type of fruit, the MRCA of *Prunus* as the D (drupe) node, and the MRCA of Rosoideae as the D/A (drupeum/achenetum) node (Figure 1A).

The results showed that the gene families underwent significant expansion and contraction in the Rosaceae family. In the Maleae tribe, the number of gene family expansions was significantly greater than the number of contractions, with the average number of expansions being 2355, which was 1.53 times the number of contractions, except for *E. japonica*. In contrast, the average number of gene family expansions in *Prunus* and Rosoideae was 1175.5, which was 0.89 times the number of contractions (Figure 1A).

Specifically, the expansion of the MADS-box gene family was detected at the P, D, and D/A nodes, with gene numbers of 13, 5, and 5, respectively. These genes include multiple *AGAMOS-like* (*AGL*) genes and their homologs, as well as *myocyte enhancer factor 2* (*MEF2*)-*like* genes and their homologs. The gene ontology (GO) enrichment results showed that these genes are mainly involved in the regulation of reproductive processes and the development of reproductive structures. In the Maleae tribe, they are also involved in the regulation of flower development, meristem development, and floral organ senescence (Figure 1B–D). The correlation between the number of MADS-box genes at the P, D, and D/A nodes and their expanded functions may be related to the recent WGD event.

### 2.2. The Retention of Duplicated Genes in Maleae following WGD and Subsequent Chromosome Rearrangement

The results showed that the species in the *Prunus* genus and Rosoideae subfamily had strong collinearity and few structural variations (SV) (Appendix A), while those in the Maleae tribe had many SVs and inferior collinearity (Appendix A). To investigate the evolutionary process of chromosomes in Maleae, we reconstructed the ARK with nine ancient chromosomes and subsequently analyzed the karyotype evolution. The karyotypes of *F. × ananassa* and *P. avium* can be directly formed by seven fissions/nine fusions and seven fissions/eight fusions of the ARK, respectively (Figure 2A). Chromosomes 4 and 7 of *F. × ananassa* and chromosomes 8, 3, 5, and 2 of *P. avium* are identical to the ARK, similar to previous findings [12,40].

The Maleae tribe has undergone duplication and complex chromosome evolution. After duplication, the karyotypes of *C. pinnatifida*, *E. japonica*, *P. pyrifolia*, and *M. × domestica* experienced 28 fissions/20 fusions, 59 fissions/51 fusions, 33 fissions/25 fusions, and 29 fissions/21 fusions, respectively (Figure 2A).

The distribution of synonymous substitutions (Ks) of homologous genes showed significant peaks in all Rosaceae species at around Ks = 1.5, corresponding to the γ event that occurred in the common ancestor of core eudicots. Additionally, another peak was detected in Maleae near Ks = 0.2, corresponding to the recent WGD event, which occurred at approximately 14.69 Ma (Figure 2B), consistent with the phylogenetic inference results.

We filtered out data with gene copies present only in one species or only in the Maleae tribe, leaving 29,239 orthologous groups present in both Maleae and *Prunus* and Rosoideae. Undoubled orthologous groups indicate that the number of genes in Maleae is less than twice that of other species, accounting for 64.52% of the total. Additionally, 44.51% of these orthologous groups show no expression in the flowers, branches, fruits, leaves, and roots. Doubled orthologous groups indicate that the number of genes in Maleae is more than twice that of other species, with 8979 orthologous groups expressed in plant tissue, which is more than six times that of the unexpressed orthologous groups (1396) (Figure 2C).

### 2.3. Four MADS-Gene Orthologous Groups Are Expressed in Plant Tissues

The genes in the orthologous groups that have doubled in quantity and are expressed are involved in basic biological processes, cellular components, and molecular functions in plants, including metabolic processes, organelle parts, membrane and cytoplasmic parts, nucleic acid binding, and catalytic activity (Appendix A). To explore the functional differentiation of genes after duplication, we filtered out 2420 tissue-specific expressed genes (Figure 3A), among which four orthologous groups were related to MADS-box genes, with the numbers of these genes in Maleae being about twice those in *Prunus* and Rosoideae.

Among them, OG0000536 is an orthologous group of *MADS12*, and *MADS12* is an *AP1-like* gene involved in specifying the identity of perianth organs [41]. In this study, gene copies in OG0000536 were expressed in the flowers, branches, fruits, leaves, and roots. OG0001383 is an orthologous group of *MADS2*, with some gene copies expressed in the flowers, branches, fruits, and leaves. OG0005321 is an orthologous group of *MADS18*, which is a homologous gene of *AGL9* [42]. OG0008543 is an orthologous group of *MADS6*, which is a homologous gene of *CMB1-like* (Figure 3B). They both have two and three gene copies, respectively, in Maleae and are only expressed in the flowers and fruits.

### 2.4. The MADS18 Genes Derived from WGD in M. × domestica Have Undergone Differential Evolution

The phylogenetic analysis results of genes in OG0005321 are consistent with the whole-genome analysis results. *CpMADS18b*, *CpMADS18c*, *EjMADS18a*, *EjMADS18b*, *PpMADS18a*, *PpMADS18b*, *MdMADS18a*, and *MdMADS18b* are all derived from the WGD event. *CpMADS18a* and *CpMADS18b* are the result of gene duplication (Figure 4A).

The results of the collinearity and physical and chemical properties support that *CpMADS18c*, Ej*MADS18a*, *PpMADS18a*, and *MdMADS18a* are orthologous genes that emerged after the species differentiated (Figure 4B). These genes have an almost identical size, molecular weight (Mw), isoelectric point (pI), instability index (II), and grand average of hydropathicity (GRAVY). The *MADS18b* genes from Maleae are also orthologous genes that emerged after the species differentiated, sharing a similar gene structure. However, *MdMADS18b* has undergone significant evolution, containing only seven coding sequence (CDS) regions. The encoded protein sequence has become shorter and more unstable, with a reduced Mw and pI (Appendix A).

The identity of the protein sequences between *MdMADS18a* and *MdMADS18b* is only 85.23% (Figure 4C). The N-terminus of *MdMADS18b* specifically lacks 60 amino acids, and this polymorphic segment encodes a MEF2-like domain (Appendix A), ultimately resulting in the absence of at least one α-helix and two β-sheets in its protein structure (Appendix A).

To investigate the impact of sequence differences on their expression levels, we tested the expression of *MdMADS18a* and *MdMADS18b* in different tissue types, using the expression of *FaMADS18* and *PavMADS18* as controls for the ancestral status. The results showed that all four genes were expressed in flower and fruit tissue, but not in leaf tissue (Figure 4D).

Specifically, in the flesh tissue 30 days after pollination (DAP), the expression level of *MdMADS18b* was significantly lower than that of *MdMADS18a*. In the flesh tissue at 90 DAP, the expression level of *MdMADS18b* had decreased to zero, while the expression level of *MdMADS18a* continued to increase (Appendix A). The expression patterns of the two genes in flower organs are also different: *MdMADS18a* is almost absent in the stamen and sepal, whereas *MdMADS18b* is expressed in almost all flower organs, and the expression level of *MdMADS18b* is significantly higher than that of *MdMADS18a* (Appendix A and Figure 4E).

### 2.5. The MADS6 Genes Derived from Both WGD and Tandem Duplication in M. × domestica Exhibit Differential Evolution

The phylogenetic analysis results of genes in OG0008543 are consistent with the whole-genome analysis results (Figure 5A). Among them, the two gene copies in *C. pinnatifida*, *E. japonica*, *P. pyrifolia*, and *M. × domestica* were all derived from the WGD event. Additionally, *MADS6b* originated from the gene duplication of *MADS6a* in Maleae (Figure 5B).

The physical and chemical properties indicate that the *MADS6* genes in Maleae have undergone different evolutionary changes (Appendix A). Taking *M. × domestica* as an example, *MdMADS6a* and *MdMADS6b* are derived from gene duplication, but *MdMADS6a* has specifically deleted or mutated 27 amino acids at the C-terminus, resulting in a shorter protein sequence with a smaller Mw and pI. Compared with *MdMADS6b*, *MdMADS6c* has a specific mutation of 41 amino acids, and its encoded protein sequence length remains unchanged, but both its pI and II are reduced. The identity of the protein sequences of *MdMADS6a* and *MdMADS6b* is 85.14%, while the identity of the protein sequences of *MdMADS6c* and *MdMADS6b* is 83.53% (Figure 5C). The protein structures of the three are not significantly different in composition (Appendix A).

We investigated the expression of *MdMADS6a*, *MdMADS6b*, and *MdMADS6c* in different tissue types, using the expression of *FaMADS6* and *PavMADS6* as controls for the ancestral status. The results showed that four genes were expressed in flower and fruit tissue, but not in leaf tissue, with the exception of *PavMADS6*, which was only expressed in fruits (Figure 5D). The expression levels of *FaMADS6* and *PavMADS6* in fruits were significantly higher than those in flowers. In contrast, the expression levels of *MdMADS6a*, *MdMADS6b*, and *MdMADS6c* in fruits were lower than those in flowers.

Specifically, in the flesh tissue at 30 DAP, the expression levels of *MdMADS6a* and *MdMADS6b* were similar but lower than that of *MdMADS6c*. In the flesh tissue at 90 DAP, the expression level of *MdMADS6a* had significantly increased, while *MdMADS6b* and *MdMADS6c* were almost absent. In both petal and sepal tissue, the expression patterns of the three gene copies were similar: the expression level of *MdMADS6c* was the highest, followed by *MdMADS6b*, and the expression level of *MdMADS6a* was the lowest, with significant differences among the three genes. The expression level of *MdMADS6b* in the pistil was significantly higher than that of *MdMADS6a* and *MdMADS6c*. The expression levels of the three genes in the stamen were very low (Figure 5E and Appendix A).

## 3. Discussion

### 3.1. Changes in Chromosomes and Genes Following the WGD Event in Rosaceae

The connection between whole-genome polyploidization and angiosperm evolution has been confusing among researchers [1,4]. Many studies have confirmed that the Maleae tribe experienced a recent WGD event, while other species in Rosaceae did not [12,43]. In addition, many species in Rosaceae are closely related to humans, such as apples, peaches, and strawberries, which are important economic crops and essential sources of various nutrients, such as vitamins and dietary fiber [44,45,46]. The Rosaceae family contains abundant and easily accessible genomic resources [47]. To explore the relationship between WGD and the differentiation of the Rosaceae family, we chose the genomes of the Maleae tribe, which experienced a recent WGD event, and *Prunus* and Rosoideae, which did not.

We inferred that the recent WGD event experienced by Maleae likely occurred between 6.44–12.36 Ma and 40.10–81.09 Ma, with the specific time possibly around 14.69 Ma, through the reconstruction of the phylogenetic trees and the analysis of the species divergence times (Figure 1A and Figure 2B).

Compared with other species, the Maleae tribe not only doubled its number of chromosomes but also experienced complex chromosome breakage and rearrangement events (Figure 1A and Figure 2A). Moreover, many structural variations occurred after the differentiation of *C. pinnatifida*, *E. japonica*, *P. pyrifolia*, and *M. × domestica*, resulting in weaker collinearity compared to *Prunus*, suggesting that the Maleae are still undergoing rapid evolution (Appendix A).

Many studies have shown that multiple ancient polyploidization events and gene duplications in angiosperms have made significant contributions to their adaptation to different environments [5,6]. For example, *HSF* transcription factors, which play an important role in the plant heat stress response, rapidly expanded during ancient polyploidization, with the expansion closely related to the warm and arid climate [48,49]. The *C-repeat binding factors* (*CBFs*) *are* the main regulatory factors for cold acclimation in many angiosperm plants. All *CBF transcription* factors are derived from the duplication and functional differentiation of the same DREB III gene, and these gene duplication events were concentrated during the ice ages, when the global average temperature significantly decreased, helping plants to adapt to the global cooling [50,51,52].

We found that approximately 35.48% of the genes in Maleae originated from WGD, with the majority of genes expressed in the plant tissue and involved in basic biological processes such as the response to stimuli (Figure 2C and Appendix A).

### 3.2. The Expansion of the MADS-Box Gene Family Is Associated with Different Fruit Types in Rosaceae

The rapid diversification of angiosperms relies on the evolution of diverse floral organs and fruit types [17]. Coincidentally, the Rosaceae family includes various fruit types: pome, drupe, and drupetum/achenetum fruits. Correspondingly, Rosaceae also exhibit flower organs with significant morphological differences [15,16].

The MADS-box gene family is deeply involved in various stages of flower organ morphogenesis and fruit development. Some studies have shown that the differentiation of Arabidopsis flowers and fruits is controlled by two duplication genes, *SHATTERPROOF1* (*SHP1*) and *SHP2*. The expression of these two genes is also necessary in promoting ovule identity [28,29].

In addition, there is a reproductive strategy where seeds develop without fertilization, known as apomixis [53], which is also influenced by MADS-box genes. The overexpression of *ZaMADS80* in *Zanthoxylum armatum* triggered precocity and parthenocarpy, and the interaction between *ZaMADS80* and *ZaMADS67* (*AGL32-like*) may contribute to apomixis [54]. In the ovule of *Brachiaria brizantha*, the differential expression of *BbrizAGL6* is the main difference between sexual and apomictic reproduction. Transcripts of *BbrizAGL6* are located in the megaspore mother cells of the ovaries from both apomictic and sexually reproducing plants, as well as in the nucelli of apomictic plants, in the region where aposporic initial cells differentiate [55]. The MADS-box genes can facilitate asexual reproduction by bypassing the need for fertilization; however, there is still a lack of relevant research in Rosaceae.

We found that the MADS-box gene family members underwent expansion at the MRCA nodes of different fruit types, and the difference in the number of expansions was associated with WGD (Figure 1B–D). Subsequently, we specifically filtered the MADS-box genes that were duplicated and preserved during the WGD event and found two typical examples: the orthologous groups of *MADS18* and *MADS6* (Figure 3B).

### 3.3. MADS18 and MADS6 Are Expanded through WGD and Gene Duplication

In these two examples, the Maleae tribe contains two gene copies derived from WGD, while *Prunus* and Rosoideae mostly contain one gene copy. In addition, *C. pinnatifida* has three copies of *MADS18*, while other species in Maleae only have two copies (Figure 4A,B). Based on the phylogenetic and collinearity analysis, it is speculated that this gene duplication event only exists in *C. pinnatifida*. *C. pinnatifida* has two copies of *MADS6*, while other species in Maleae have three copies. These gene copies have undergone various sequence differentiations in different species, so it is speculated that this gene duplication event occurred in the ancestors of Maleae, underwent differentiation among species, and lost one copy in *C. pinnatifida* (Figure 5A,B).

### 3.4. MADS18 and MADS6 Have Different Expression Levels in Apple Fruits at Different Stages and Different Flower Organs

*F. × ananassa* and *P. avium* both have one copy of *MADS18* and *MADS6* and have not experienced WGD, so their expression is considered to represent the Maleae ancestors before the WGD. All genes were expressed in both mixed fruit and mixed flower tissue. These genes were not expressed in leaf tissue, which served as a negative control, and transcriptome data showed that they were not expressed in the roots and branches but were specifically expressed in the flowers and fruits (Figure 4D and Figure 5D).

We found that sequence differentiation occurs between *MdMADS18a* and *MdMADS18b*, as well as among *MdMADS6a*, *MdMADS6b*, and *MdMADS6c*, which affects their physical and chemical properties and protein structures (Figure 4C and Figure 5C, Appendix A). We divided the fruit samples into two stages, 30 DAP and 90 DAP, and divided the flower organs into the petal, pistil, stamen, and sepal. Surprisingly, the expression patterns of these genes have undergone significant changes in a more detailed structure, with some even no longer expressed (Figure 4E and Figure 5E). These genes are likely to play a crucial role in the evolution of the fruit types and floral organs of pomes.

## 4. Materials and Methods

### 4.1. Collection of Publicly Available Genomic and Transcriptome Data

We collected genomic and gene annotation information that is public, including that on *Crataegus pinnatifida* var. *major* N.E. Br., *Eriobotrya japonica* (Thunb.) Lindl., *Pyrus pyrifolia* Nakai, *Malus × domestica* Borkh., *Prunus avium* L., *Prunus persica* Batsch, *Prunus armeniaca* L., *Prunus salicina* Lindl., *Fragaria × ananassa* Duchesne ex Rozier, *Rubus argutus* Link, *Rubus occidentalis* L., *Rubus idaeus* L., and *Vitis vinifera* L. (Appendix A). In addition, we downloaded the raw transcriptome sequence data for different plant tissue types, including the flower, leaf, root, fruit, and branch from *M. × domestica* (Appendix A).

### 4.2. Phylogenetic Analysis and Time Inference

A total of 351 single-copy orthologous genes were selected from 13 species using OrthoFinder2 (https://github.com/davidemms/OrthoFinder, accessed on 29 October 2021) [56], with *V. vinifera* as the outgroup. All protein sequences were aligned with MUSCLE (https://www.drive5.com/muscle, accessed on 15 August 2021) [57] and Gblocks (https://www.biologiaevolutiva.org/jcastresana/Gblocks.html, accessed on 15 August 2021) [58]. All sites including missing and gap data were used to construct the phylogenetic tree with the maximum likelihood method using RAxML (https://github.com/stamatak/standard-RAxML/archive/master.zip, accessed on 16 August 2021) [59], applying the PROTGAMMAJTTF model and 1000 bootstraps.

The divergence times were estimated based on the phylogenetic tree using an approximate likelihood method implemented in MCMCtree of the PAML 4.9 package (http://abacus.gene.ucl.ac.uk/software/paml4.9j.tgz, accessed on 10 August 2022) [60] with the options “normal approximation” and the “JC69” model, using the parameters “burnin = 400,000; sampfreq = 10; nsample = 100,000”.

### 4.3. GO Term Enrichment

We extracted sequence IDs using in-house Python (https://www.python.org/, accessed on 16 March 2020) scripts and sorted and deduplicated them. Then, we submitted them to the HortGenome Search Engine (http://hort.moilab.net/#/, accessed on 15 July 2024) [61]. The result was visualized using R scripts (https://www.r-project.org/, accessed on 16 March 2020).

### 4.4. Gene Family Expansion and Contraction

The orthologous group analysis was performed using OrthoFinder2 and the phylogenetic tree as mentioned above. Expansions and contractions of all 13 species were detected using CAFÉ (https://github.com/hahnlab/CAFE5.git, accessed on 10 July 2024) [62] with the default parameters. The expanded and contracted gene families were subjected to statistical analysis and visualization using Excel (https://www.microsoftstore.com.cn/software/office, accessed on 25 April 2019).

### 4.5. Collinearity and Sequence Similarity Analysis

Whole-genome collinearity analysis was performed with JCVI (https://github.com/tanghaibao/jcvi/wiki/MCscan-(Python-version), accessed on 15 October 2020). Credible collinear blocks were obtained using GFF files and protein sequence files, filtered with the parameter “--cscore=.99”, and visualized with the parameter “-m jcvi.graphics.karyotype”, and segments of chromosomes were visualized with the parameter “-m jcvi.graphics.synteny”.

Protein sequence alignment and similarity analysis were performed with MUSCLE as mentioned above. The visualization of gene structures was performed using TBtools (https://github.com/CJ-Chen/TBtools-II/releases, accessed on 15 March 2024) [63].

### 4.6. Ancestral Chromosome Karyotype Reconstruction and Evolution

We reconstructed the ancestral Rosaceae karyotype (ARK), and the standard process of WGDI (https://github.com/SunPengChuan/wgdi, accessed on 2 March 2023) [64] was used to infer karyotypes. First, BLASTP (https://ftp.ncbi.nlm.nih.gov/blast/executables/blast+/LATEST/, accessed on 28 August 2020) was used to search for homologs, with the parameter “-evalue 1e-5 -max_target_seqs 5 -outfmt 6”. Second, the shared “synteny blocks” with telomeres (chromosome-like) and intact chromosomes across extant genomes were explored. Third, shared blocks, including small syntenic blocks, were deleted and the remaining parts were merged into an ancestor length file, using the parameter “-ak” to produce an ancestor annotation file. Fourth, an additional round of “synteny exploration” was performed for each extant genome to extract all protochromosomes until there were no genomic blocks left, following the parameter “-icl”, “-bi” and “-d”.

### 4.7. Analysis of WGD and Gene Duplication

WGDI was used to infer duplication events, the parameter “-icl” was used to classify homologs, the parameter “-ks” was used to calculate the Ks between homologs, the parameter “-bi” was used to merge the collinearity message and Ks values, the parameter “-bk” was used to visualize the results, and the parameters “-kp” and “-pf” were used to fit the homologs generated by the same duplication event into a peak. The time at which the WGD events occurred was calculated using “T = Ks/2r” (where “T” represents the time in millions of years and “r” represents the nucleotide replacement rate).

We used MCScanX (https://github.com/wyp1125/MCScanX, accessed on 15 August 2022) [65] to classify the duplication events into WGD or segmental (WGD), tandem (TD), and proximal duplications (PD).

### 4.8. Transcriptome Analysis

We verified the completeness and correctness of the downloaded raw transcriptome data following the parameter “vdb-validate”. We aligned the RNA-seq data to the reference genomes using HISAT2 (https://daehwankimlab.github.io/hisat2/, 20 August 2022) [66] and calculated the expression level for each gene using StringTie (https://github.com/gpertea/stringtie, 20 August 2022) [66].

### 4.9. Conserved Domain Search and Protein Structure Prediction

All protein sequences of MADS18 and MADS6 were submitted to the conserved domain search service of NCBI to identify the conserved domains (https://www.ncbi.nlm.nih.gov/Structure/cdd/wrpsb.cgi, accessed on 19 July 2024).

The protein sequences of *MdMADS18a*, *MdMADS18b*, *MdMADS6a*, *MdMADS6b*, and *MdMADS6c* were submitted to the AlphaFold Server (https://golgi.sandbox.google.com/, accessed on 19 July 2024) [67] generate highly accurate protein structures.

### 4.10. Plant Material and qRT-PCR

The tested *M. × domestica* plants were grown at the Shangzhuang experimental station of China Agricultural University in Beijing. Fresh leaves, fresh fruits at 30 DAP and 90 DAP, and fresh flowers at the late pink bud stage were collected. The tested *P. avium* plants were grown at the Beijing Academy of Agriculture and Forestry Sciences in Beijing. Fresh leaves, fresh fruits at 30 DAP, and fresh flowers were collected. The tested *F. × ananassa* plants were grown in the greenhouse in China Agricultural University in Beijing. Fresh leaves, fresh fruits at 30 DAP, and fresh flowers were collected. The RNA of tissue was extracted using the Promega RNG extraction kit (Promega Biotech, Beijing, China). cDNA was synthesized from 0.5 μg of total RNA using the ReverTra Ace™ qPCR RT Master Mix with gDNA Remover (Toyobo, Japan).

qRT-PCR was performed in a 7500 Real-Time PCR System (Applied Biosystems, Foster City, CA, USA) using TB Green Premix Ex Taq^™^ (TaKaRa, Dalian, China), according to the manufacturer’s instructions. The primers were designed using NCBI Primer BLAST (https://www.ncbi.nlm.nih.gov/tools/primer-blast/, 19 July 2024) under the following principles: specific primers were designed in the non-conserved regions of each sequence, with a length of 18–25 bp and a Tm value of around 60 °C, avoiding secondary structures and repetitive sequences. The product length was approximately 80–200 bp. For each sequence, 3–5 pairs of primers were designed, and their effectiveness was assessed by agarose gel electrophoresis. Finally, an optimal pair of primers was selected. The experiments were carried out in a total volume of 20 μL, containing 10 μL of 2× SYBR Green qPCR Mix (Beijing, China), 0.1 μM of each specific primer, and 100 ng of template cDNA. The reaction mixtures were heated to 95 °C for 10 min, followed by 40 cycles of 95 °C for 10 s and 60 °C for 32 s, 95 °C for 15 s, 60 °C for 60 s, increased by 0.3 °C each time. We measured the melt curve until the end. Three biological replicates were performed for each sample and normalized using the *actin* gene as an internal control. The 2^−ΔΔCt^ method [68] was used to calculate the relative expression levels. Detailed information about the *actin* gene and primers is shown in (Appendix A).

## 5. Conclusions

In this study, we selected the genomes of Rosaceae species from before and after WGD and focused on the differences in the fruit types and flower organs. We reconstructed the phylogenetic tree and the ancestral Rosaceae karyotype (ARK) with nine ancient chromosomes. The results revealed that the Maleae tribe has continuously experienced chromosome breakage and rearrangement following the WGD, and it was inferred that this event occurred at approximately 14.69 Ma. We identified approximately 30.71% of genes that underwent duplication and functional differentiation following the WGD. Among these, the orthologous groups of MADS18 and MADS6 contained both WGD and single gene duplication copies. These copies underwent sequence differential evolution and exhibited significant differential expression in apple fruits at different developmental stages and in different floral organs, providing new insights into and solid evidence for the impact of WGD on species evolution.

## Figures and Tables

**Figure 1 ijms-25-08962-f001:**
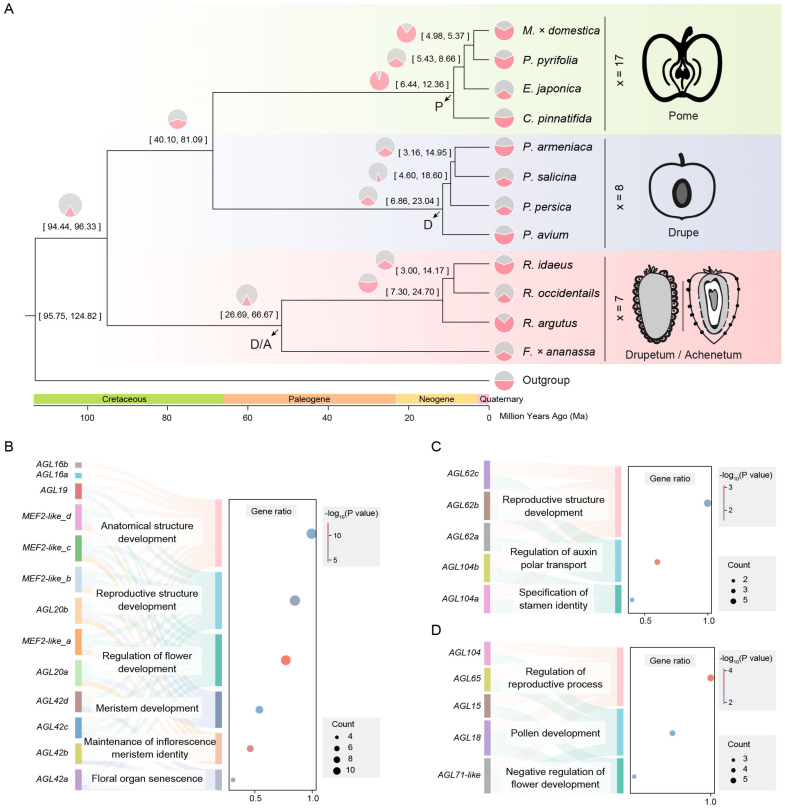
Expansion analysis of MADS-box gene family in different fruit types of Rosaceae. (**A**) The phylogenetic analysis of 12 Rosaceae species uses *V. vinifera* as the outgroup. The Rosaceae family is divided into drupetum/achenetum (pink), drupe (light blue), and pome (light green). The numbers at the branch node indicate the divergence time with a 95% confidence interval. The pie charts show the relative sizes of expansion (light pink) and contraction (light gray) of the gene family. We define the MRCA of pome fruits as the P node, the MRCA of drupe fruits as the D node, and the MRCA of drupeum/achenetum fruits as the D/A node. (**B**) GO enrichment results of expanded MADS-box genes at the P node. The lowercase suffix letters represent gene copies within the same orthologous group (the same below). (**C**) GO enrichment results of expanded MADS-box genes at the D node. (**D**) GO enrichment results of expanded MADS-box genes at the D/A node.

**Figure 2 ijms-25-08962-f002:**
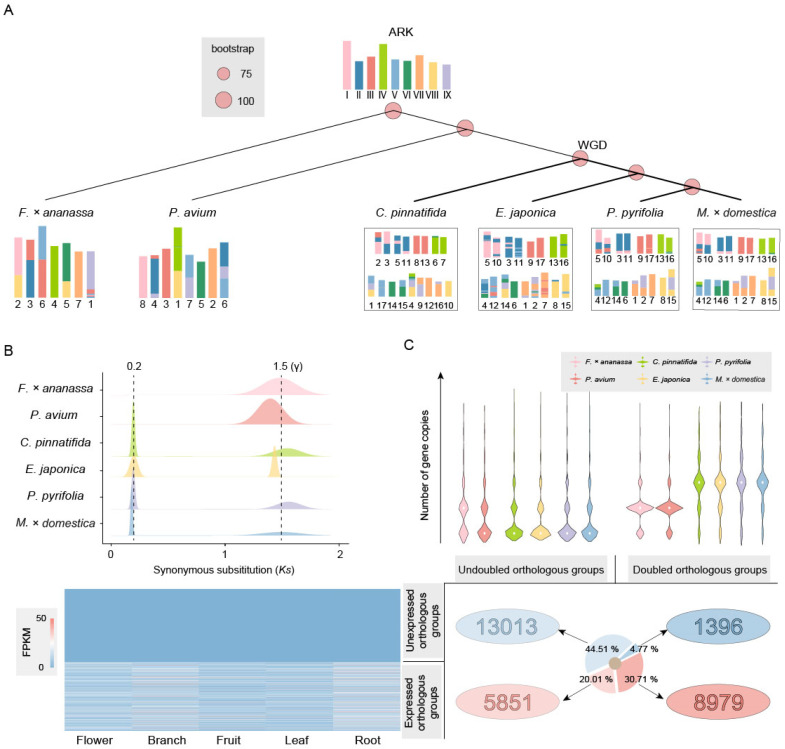
Chromosome karyotype and gene evolution after WGD in Rosaceae. (**A**) Evolutionary scenarios of chromosome karyotype changes between the ARK and *F. × ananassa*, *P. avium*, *C. pinnatifida*, *E. japonica*, *P. pyrifolia*, and *M. × domestica*. (**B**) Ks peaks of homologous genes in *F. × ananassa*, *P. avium*, *C. pinnatifida*, *E. japonica*, *P. pyrifolia*, and *M. × domestica*. A Ks value of 1.5 corresponds to the γ event that occurred in the common ancestor of core eudicots, about 117 ± 1 Ma. The peak at Ks = 0.2 corresponds to the recent WGD event, which occurred at approximately 14.69 Ma. (**C**) Preservation and expression of gene copies after WGD. All genes are divided into four orthologous groups: those with an undoubled number of genes not expressed in tissue, those with an undoubled number of genes expressed in tissue, those with a doubled number of genes not expressed in tissue, and those with a doubled number of genes expressed in tissue.

**Figure 3 ijms-25-08962-f003:**
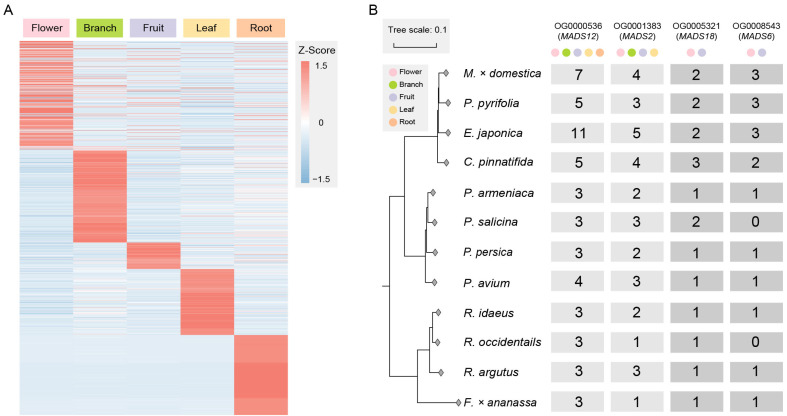
Exploring tissue-specific expressed genes and MADS-gene orthologous groups. (**A**) Tissue-specific expressed genes in doubled orthologous groups include those expressed in tissue types such as flowers, branches, fruits, leaves, and roots. (**B**) There are four orthologous groups of MADS-box genes. The number of gene copies shows a fold relationship between Maleae and other species in Rosaceae. Some gene copies from *MADS12* and *MADS2* are expressed in multiple tissue types, while the gene copies from *MADS18* and *MADS6* are only expressed in the flowers and fruits.

**Figure 4 ijms-25-08962-f004:**
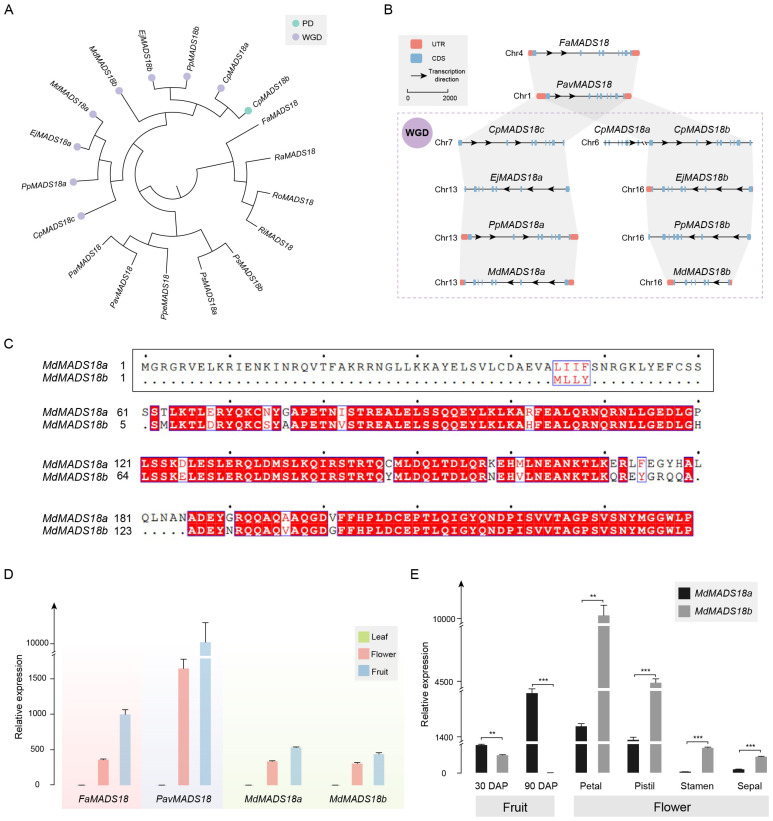
Analysis of *MADS18* orthologous group. (**A**) Phylogenetic analysis of *MADS18* gene copies in Rosaceae. The duplication events are classified into WGD or segmental (WGD) and proximal duplications (PD). (**B**) In the collinearity analysis of *MADS18* gene copies, rectangles represent gene structures, arrows indicate transcription directions, the reference line represents a length of 2000 bp, and the purple dashed box represents species that have undergone WGD. (**C**) Analysis of protein sequence similarity between *MdMADS18a* and *MdMADS18b*; the N-terminus of *MdMADS18b* specifically lacks 60 amino acids. The black border represents the sequence with a specific deletion at the beginning of *MdMADS18a* in this article. The blue border and red font indicate that more than 70% of the amino acids at this position have similar physico-chemical properties. The red background with white font indicates that the amino acids at this position are highly conserved. The black font represents standard amino acids, and the center point represents the sequence position, with each point indicating the next 10 amino acids. (**D**) Relative expression of the *MADS18* gene copies in the leaf, flower, and fruit by qRT-PCR. (**E**) Relative expression of *MdMADS18a* and *MdMADS18b* genes in fruits at different developmental stages and in different flower organs. Statistical significance was determined using Student’s *t*-test, where ** indicates *p* < 0.01, and *** indicates *p* < 0.001.

**Figure 5 ijms-25-08962-f005:**
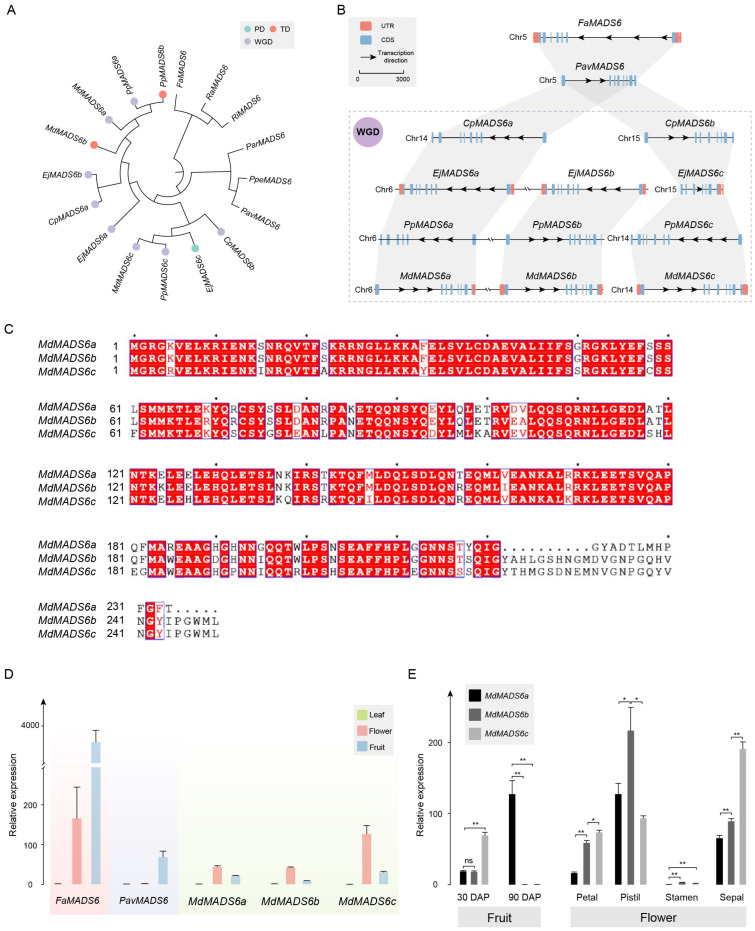
Analysis of *MADS6* orthologous group. (**A**) Phylogenetic analysis of *MADS6* gene copies in Rosaceae. The duplication events are classified into WGD or segmental (WGD), tandem (TD), and proximal duplications (PD). (**B**) In the collinearity analysis of *MADS6* gene copies, rectangles represent gene structures, arrows indicate transcription directions, the reference line represents a length of 2000 bp, and the purple dashed box represents species that have undergone WGD. (**C**) Analysis of protein sequence similarity among *MdMADS6a*, *MdMADS6b*, and *MdMADS6c*. Compared to *MdMADS6b*, *MdMADS6a* has specifically deleted or mutated 27 amino acids at the C-terminus, while *MdMADS6c* has a specific mutation of 41 amino acids. The blue border and red font indicate that more than 70% of the amino acids at this position have similar physico-chemical properties. The red background with white font indicates that the amino acids at this position are highly conserved. The black font represents standard amino acids, and the center point represents the sequence position, with each point indicating the next 10 amino acids. (**D**) Relative expression of the *MdMADS6* gene copies in the leaf, flower, and fruit by qRT-PCR. (**E**) Relative expression of *MdMADS6a* and *MdMADS6b* genes in fruits at different developmental stages and in different flower organs. Statistical significance was determined using Student’s *t*-test, where * indicates *p* < 0.05 and ** indicates *p* < 0.01.

## Data Availability

The data presented in this study are openly available in FigShare at 10.6084/m9.figshare.26425204, reference number https://doi.org/10.6084/m9.figshare.6025748, accessed on 1 August 2024.

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
