# Peer review of "Gene Duplication and Functional Diversification of MADS-Box Genes in Malus × domestica following WGD: Implications for Fruit Type and Floral Organ Evolution"

_ijms, 2024, doi:10.3390/ijms25168962_

Round 1

Reviewer 1 Report

Comments and Suggestions for Authors

In the present study, the authors have reconstructed the phylogenetic tree and the ancestral Rosaceae karyotype (ARK) from four species from the Maleae tribe and eight species from the Prunus genus and the Rosoideae. The result indicates that the Maleae tribe has continuously experienced chromosome breakage and rearrangement following the WGD and identified approximately 30.71% of genes underwent duplication and functional differentiation following WGD. The results also found that different fruit types are associated with varying copies of MADS-box gene family duplications revealing their implications for fruit type and floral organ evolution.

General comments:

1. For performant a comparative genomic analysis of MADS-box genes in Rosaceae, why did the authors include grapevine Vitis vinifera (belongs to Vitis genus/Vitaceae) which is listed in Table S1 since it is not Rosaceae? Please provide a reason for including Vitis vinifera as an outgroup.

2. Please note that some analyses are run via accessing databases or online websites. Therefore, some references were cited in “4. Materials and Methods” which showed and gave guidance for how to analyze and generate data based on browsing/online databases or software. Thus, these citations might be presented and provided with access links instead of published papers.

3. Provide the accession number /GeneBank of the gene listed in Table S4. Italicize the gene name (gene ID) in this table. In addition, the authors should describe how the primers used for qRT-PCR are designed based on the accession number /GeneBank of genes via which method or program.

4. The procedure of qRT-PCR should be represented in more detail.

Minor comments:

- Italicize the scientific name of plants “C. pinnatiida, E. japonica, P. pyrifolia, M. × domestica, P. avium, P. persica, P. armeniaca, P. salicina, F. × ananassa, R. argutus, R. occidentalis, R. idaeus, and V. vinifera” (section 4.1).

- About citation:

+ Ref 56 should be cited right after “MUSCLE (v3.8.31)” (section 4.2)

MUSCLE (v3.8.31) and Gblocks (v0.91b) [56,57]. -> “MUSCLE (v3.8.31) [56] and Gblocks (v0.91b) [57].”

+ Ref 66 cites right after “AlphaFold Server” (section 4.9):

the AlphaFold Server generate highly accurate protein structure [66].”-> “the AlphaFold Server [66] generate highly accurate protein structure.”

- Capitalize Each Word in the subsection according to the journal’s style. Especially, all subsections of section "4. Materials and Methods".

For example, section “4.1. Collection of publicly available genomic and transcriptome data”

-> “4.1. Collection of Publicly Available Genomic and Transcriptome Data”

- Rewrite “30 days after pollination (30 DAP)” (section 2.4) -> “30 days after pollination (DAP)”.

-  Italicize the gene name: “AGAMOS-like (AGL)” (section 2.1), “AP1” (section 2.3), “MADS18a and MADS18b” (section 2.4 and Figure 4’s legend), MADS6a/b/c, (section 2.5),  “actin” (section 4.10). etc. Please carefully double-check elsewhere in the whole manuscript.

- Reformat the numbered style of cited Ref. For example, [15,20,21] -> [15,20,21]

I have marked the above comments on the manuscript. Please check.

Author Response

Comments 1: For performant a comparative genomic analysis of MADS-box genes in Rosaceae, why did the authors include grapevine Vitis vinifera (belongs to Vitis genus/Vitaceae) which is listed in Table S1 since it is not Rosaceae? Please provide a reason for including Vitis vinifera as an outgroup.

Response 1: Thank you for your insightful question! The Vitaceae family is a basal group of the rosids branch of angiosperms and is clearly an older group than the species in the Rosaceae family. Using Vitaceae as the root of the phylogenetic tree can yield reliable clustering results for species within the Rosaceae family. In addition, the evolutionary research on the Vitaceae family has a solid foundation, which is beneficial for calculating the divergence time of the entire phylogenetic tree. From a personal perspective, grapes are a delicious fleshy fruit that aligns with our research topic. All these factors are why we selected Vitis vinifera as the outgroup.

Comments 2: Please note that some analyses are run via accessing databases or online websites. Therefore, some references were cited in “4. Materials and Methods” which showed and gave guidance for how to analyze and generate data based on browsing/online databases or software. Thus, these citations might be presented and provided with access links instead of published papers.

Response 2: Thank you for your advice! We have added the access links in Section 4 and highlighted it.

Comments 3: Provide the accession number /GeneBank of the gene listed in Table S4. Italicize the gene name (gene ID) in this table. In addition, the authors should describe how the primers used for qRT-PCR are designed based on the accession number /GeneBank of genes via which method or program.

Response 3: Thank you for your advice! Our gene sequences are sourced from the Genome Database for Rosaceae (https://www.rosaceae.org/), which contains more detailed data than NCBI, so we provide the original ID of genes in the genome. All gene IDs in tables have been italicized. In addition, qRT-PCR primers were designed using NCBI Primer BLAST (https://www.ncbi.nlm.nih.gov/tools/primerblast/), we have added details in Section 4.10 and highlighted.

Comments 4: The procedure of qRT-PCR should be represented in more detail.

Response 4: Thank you for your advice! We have added more details on qRT-PCR and highlighted them in Section 4.10.

Comments 5: Italicize the scientific name of plants “C. pinnatiida, E. japonica, P. pyrifolia, M. × domestica, P. avium, P. persica, P. armeniaca, P. salicina, F. × ananassa, R. argutus, R. occidentalis, R. idaeus, and V. vinifera” (section 4.1).

Response 5: Thank you for your timely reminder! This issue has been corrected and highlighted in Section 4.1.

Comments 6: About citation:

+ Ref 56 should be cited right after “MUSCLE (v3.8.31)” (section 4.2)

“MUSCLE (v3.8.31) and Gblocks (v0.91b) [56,57]. -> “MUSCLE (v3.8.31) [56] and Gblocks (v0.91b) [57].”

+ Ref 66 cites right after “AlphaFold Server” (section 4.9):

“the AlphaFold Server generate highly accurate protein structure [66].”-> “the AlphaFold Server [66] generate highly accurate protein structure.”

Response 6: Thank you for your timely reminder! We have carefully reviewed the entire manuscript and corrected this issue.

Comments 7: Capitalize Each Word in the subsection according to the journal’s style. Especially, all subsections of section "4. Materials and Methods".

For example, section “4.1. Collection of publicly available genomic and transcriptome data”

-> “4.1. Collection of Publicly Available Genomic and Transcriptome Data”

Response 7: Thank you for your timely reminder! We have carefully reviewed the Section 4 and corrected this issue.

Comments 8: Rewrite “30 days after pollination (30 DAP)” (section 2.4) -> “30 days after pollination (DAP)”.

Response 8: Thank you for your timely reminder! We have corrected this issue.

Comments 9: Italicize the gene name: “AGAMOS-like (AGL)” (section 2.1), “AP1” (section 2.3), “MADS18a and MADS18b” (section 2.4 and Figure 4’s legend), MADS6a/b/c, (section 2.5), “actin” (section 4.10). etc. Please carefully double-check elsewhere in the whole manuscript.

Response 9: Thank you for your timely reminder! We have carefully reviewed the text, figures, and tables throughout the entire manuscript, corrected the gene name issues, and made appropriate abbreviations. All corrected parts have been highlighted.

Comments 10: Reformat the numbered style of cited Ref. For example, [15,20,21] -> [15,20,21]

Response 10: Thank you for your timely reminder! We have carefully reviewed the entire manuscript and corrected the cited issues.

Reviewer 2 Report

Comments and Suggestions for Authors

The article by Baoan Wang et al. deals with the diversification of MADS-Box genes in M. × domestica. 

In terms of subject matter, the article is interesting and fits the profile of IJMS. 

The structure of the article is clear, as is the language. 

Methodologically, the article is correct.

In Results. When the authors first mention taxa, they should be written as whole names and not abbreviations. In addition, Latin names should be accompanied by the names of the authors of these species.

In Materials and Methods, the full Latin names of the species studied should be given. Please give, also the authors of these species. Generic and species names should be in italics.

How the phylogenetic analysis of 12 Rosaceae that the authors presented relates to other phylogenetic trees from other works (see https://www.ncbi.nlm.nih.gov/pmc/articles/PMC5400374/ and https://www.frontiersin.org/journals/plant-science/articles/10.3389/fpls.2024.1367645/full)

The genera in the family Rosaceae are undergoing thriving evolution, mainly through hybridization and the emergence of new hybrid taxa (including the genera Malus, Sorbus and others) but the huge role of apomixis should be emphasized. Could the authors discuss their results precisely in terms of the apomixis phenomenon.

Author Response

Comments 1: In Results. When the authors first mention taxa, they should be written as whole names and not abbreviations. In addition, Latin names should be accompanied by the names of the authors of these species. In Materials and Methods, the full Latin names of the species studied should be given. Please give, also the authors of these species. Generic and species names should be in italics.

Response 1: Thank you for your timely reminder! We have replaced the abbreviations with full names in Section 2.1, provided the full Latin names of all species, including the authors, and italicized the generic and species names. All corrected parts have been highlighted.

Comments 2: How the phylogenetic analysis of 12 Rosaceae that the authors presented relates to other phylogenetic trees from other works (see https://www.ncbi.nlm.nih.gov/pmc/articles/PMC5400374/ and https://www.frontiersin.org/journals/plant-science/articles/10.3389/fpls.2024.1367645/full)

Response 2: Thank you for your insightful question! We selected single-copy nuclear genes, and the first article (Xiang et al., 2017) also used low-copy orthologous and single-copy nuclear genes for multiple analyses. Our results are consistent, and the structure of the phylogenetic tree is clear: the Maleae species are clustered together, the Amygdaleae species (including the Prunus genus) are clustered together, all belonging to the Amygdaloideae subfamily; the Rubeae species are clustered together, along with Potentilleae (including the Fragaria genus), belonging to the Rosoideae subfamily. The second article (Sun et al., 2024) analyzed the chloroplast genome and obtained detailed phylogenetic relationships within the core Maleae, with Pyrus pyrifolia and Eriobotrya japonica showing closer clustering relationships, which may be related to the different datasets. The chloroplast genome is inherited through the maternal lineage, and the nuclear genome can better represent the overall evolutionary history, especially in the context of hybridization and gene flow. Additionally, the Malus and Pyrus genera can produce hybrid offspring and should have a closer genetic relationship, making our results and those of Xiang et al. more representative.

Comments 3: The genera in the family Rosaceae are undergoing thriving evolution, mainly through hybridization and the emergence of new hybrid taxa (including the genera Malus, Sorbus and others) but the huge role of apomixis should be emphasized. Could the authors discuss their results precisely in terms of the apomixis phenomenon.

Response 3: Thank you for your insightful question! Apomixis bypasses the normal meiosis and fertilization processes to directly form embryos and produce seeds, and it is present in many flowering plants, including the Potentilleae tribe of the Rosaceae family. Our study found that MADS-box genes undergo sequence and functional differentiation after duplication, while apomixis is linked to polyploidy, which may potentially be related to the differentiation of apomixis. For example, in ovules of Brachiaria brizantha, where the main differences between sexual and apomictic reproduction occur, BbrizAGL6 was differentially modulated.

Direct research on the relationship between MADS-box genes and apomixis in the Rosaceae family is limited. To learn more about the specific relationship between them, it would be best to conduct detailed experiments on materials of the same species that undergo sexual and asexual reproduction separately, including phenotypic differences such as meiosis, double fertilization, and endosperm development, as well as investigating the specific mechanisms of related regulatory genes.

In addition, we have added the discussed content to Section 3.2 and highlighted it.